Spawning aggregations of checkered snapper (Lutjanus decussatus) and blackspot snapper (L. fulviflamma): seasonality, lunar-phase periodicity and spatial distribution within spawning ground

Nanami Atsushi nanami_atsushi71@fra.go.jp nanami@fra.affrc.go.jp
Yaeyama Field Station, Coastal and Inland Fisheries Ecosystem Division, Environment and Fisheries Applied Techniques Research Department, Japan Fisheries Research and Education Agency , Ishigaki , Okinawa , Japan
Esteban María Ángeles
Electronic publication date: 2023 Sep 11
Publication date: 2023
Volume: 11
Electronic Location ID: e15991
Received 2023 Apr 27; Accepted 2023 Aug 8
Copyright: ©2023 Nanami
Copyright year: 2023
Copyright holder: Nanami
License: This is an open access article distributed under the terms of the Creative Commons Attribution License, which permits unrestricted use, distribution, reproduction and adaptation in any medium and for any purpose provided that it is properly attributed. For attribution, the original author(s), title, publication source (PeerJ) and either DOI or URL of the article must be cited.
License URL: https://creativecommons.org/licenses/by/4.0/

Keywords: Spawning aggregation, Snapper, Lutjanus decussatus, Lutjanus fulviflamma, Lunar periodicity, Seasonality, Gonad development, Spawning ground

Funding: Environment Research and Technology Development Fund of the Ministry of the Environment, Japan S-15-3(4): JPMEERF16S11513 This study was supported by the Environment Research and Technology Development Fund (S-15-3(4): JPMEERF16S11513) of the Ministry of the Environment, Japan. The funders had no role in study design, data collection and analysis, decision to publish, or preparation of the manuscript.

==============================
Snappers (family Lutjanidae) are important fisheries target species and some species are known to form spawning aggregations at particular spawning grounds. The present study investigated the ecological characteristics of fish aggregations of two snapper species (checkered snapper Lutjanus decussatus and blackspot snapper L. fulviflamma) that form at a particular site. Specifically, the aims were to clarify (1) seasonality and lunar-phase periodicity of fish aggregation formation, (2) fine-scale spatial distribution of fish density (spatial variations of fish density at intervals of several-tens meters) within the aggregation site, (3) size and age frequency distributions of fishes in the aggregation site, (4) gonad development, (5) to compare fish abundance between inside and outside the aggregation site, and (6) to verify that fish aggregations of the two snapper species were spawning aggregation. Underwater observations using a 600 m × 5 m transect revealed that greater fish abundance of Lutjanus decussatus was found monthly between May and October, and clear positive peaks in the fish abundance were found only around the last-quarter moon. This lunar-related periodicity in the increase of fish abundance was confirmed by a time-series analysis (correlogram). Within the aggregation site, L. decussatus showed a relatively uniform distribution. In contrast, greater fish abundance of L. fulviflamma was found monthly between April and October, and clear positive peaks in the fish abundance were found around the last-quarter moon (April, May, June and October) or new moon (July, August and September). This lunar-related periodicity was also confirmed by correlogram. Lutjanus fulviflamma showed a relatively clumped distribution within the aggregation site. Most females of the two species in the aggregation site had hydrated eggs, indicating that the two species form aggregations for reproduction. The two species, although occurring simultaneously, are considered to form aggregations of conspecifics only. For L. decussatus, average fork length and age of males and females were 229.2 mm and 243.9 mm and 9.4 years and 8.1 years, respectively. For L. fulviflamma, average fork length and age of males and females were 233.9 mm and 246.9 mm and 6.8 years and 8.1 years, respectively. Fish abundance inside the aggregation site was 266.8-fold and 141557.1-fold greater than those outside the aggregation site for L. decussatus and L. fulviflamma, respectively. These results showed that (1) fish aggregation formation of the two snapper species was predictably repeated in particular months and lunar-phase, (2) it was predictably found at the particular site, (3) the fish abundance in the aggregation site markedly exceeded the fish abundance outside the aggregation site, and (4) the two species form aggregations for reproduction. Therefore, it is suggested that the fish aggregations for the two species can be regarded as spawning aggregations.

Introduction

Coral reef fishes show diverse reproductive behaviors (Thresher, 1984). Among these diverse fish species, some fish species form aggregations with greater densities during restricted seasons and lunar phases at particular spawning grounds (Nemeth, 2009). These fish aggregations are known as spawning aggregations (Sadovy de Mitcheson & Colin, 2012). Domeier (2012) defined spawning aggregations as conspecific individuals gather at specific sites in a specific period. Over 80 fish species are regarded to form spawning aggregations (Sadovy de Mitcheson & Colin, 2012), although it is suggested that the actual number of species would be greater than 160 (Claydon, 2004). Two types of spawning aggregations have been reported: resident and transient spawning aggregations (Domeier & Colin, 1997). Resident spawning aggregations are characterized by smaller-sized species (e.g., parrotfishes, surgeonfishes and wrasses), shorter migration distance (within several kilometer scale), shorter duration of the spawning event (several hours) and almost daily aggregation. On the other hand, transient spawning aggregations are characterized by larger-sized species (e.g., emperorfishes, groupers and snappers), longer migration distances (scales of several to several-hundreds of kilometers), longer duration of the spawning event (several days to several weeks) and almost monthly or annual aggregations (Nemeth, 2009).

Some fish species that form transient spawning aggregations include larger-sized fishery target species such as emperorfishes, groupers and snappers (Sadovy de Mitcheson & Colin, 2012). These fish species are mesopredators, which have significant roles in coral reef ecosystems, that control population size of smaller-sized fishes belonging to lower trophic levels (Graham, Evans & Russ, 2003). Hence, effective protection of the spawning aggregations for these species is needed (Beets & Friedlander, 1999; Linderman et al., 2000; Sala, Ballesteros & Starr, 2001; Nemeth, 2005; Sadovy & Domeier, 2005; Russell, Luckhurst & Lindeman, 2012), since formation of transient spawning aggregations is spatially and temporally predictable and such spawning aggregations have great vulnerability to fishing (Samoilys, 1997; Rhodes & Tupper, 2008; Sadovy de Mitcheson et al., 2008; de Mitcheson & Erisman, 2012; Chollett et al., 2020; Pittman & Heyman, 2020).

Snappers (family Lutjanidae) are important fishery target species and mesopredators in tropical and sub-tropical waters (Allen, 1985; Polovina & Ralston, 1987; Nanami & Shimose, 2013; Taylor et al., 2018; Amorim et al., 2019; Menezes et al., 2022) and at least 12 species are regarded to form transient spawning aggregations (Sadovy de Mitcheson & Colin, 2012). Some ecological characteristics of snappers in terms of spawning aggregations have been studied such as location of spawning ground (Claro & Lindeman, 2003; Heyman & Kjerfve, 2008; Sakaue et al., 2016; Malafaia, França & Olavo, 2021), seasonality and lunar-phase periodicity of spawning aggregation formation (Kadison et al., 2006; Sakaue et al., 2016; Biggs & Nemeth, 2014; Biggs & Nemeth, 2016; Cimino et al., 2018), spawning migration movements (Farmer & Ault, 2011; Feeley et al., 2018) and spawning behavior (Carter & Perrine, 1994; Heyman et al., 2005; Sadovy de Mitcheson, Colin & Sakaue, 2012).

Two snapper species, checkered snapper Lutjanus decussatus (Cuvier, 1828) and blackspot snapper L. fulviflamma (Forsskål, 1775), are important fisheries target species in Okinawan coral reefs (Akita et al., 2016). Nanami et al. (2010) indicated the possibility that L. decussatus forms spawning aggregations in this region by clarifying its reproductive biology. The main spawning season was between June and October and clear lunar-synchronized fluctuations in the gonadsomatic index (GSI) of females were found. The highest GSI values were found around the last-quarter moon phase during the main spawning season. However, direct evidence (fish aggregation at a particular site) has not yet been reported. In contrast, Shimose & Nanami (2015) showed that the main spawning season of L. fulviflamma occurred between April and August. However, lunar-synchronized reproductive activity as well as the possibility of spawning aggregation formation of L. fulviflamma have not been examined yet. Although there has been no ecological information from the local communities about spawning aggregations of the two species, fish aggregations of the two snapper species were found in an Okinawan coral reef (Fig. 1, Videos S1 and S2). This finding suggests that the fish aggregation might be a spawning aggregation and should be appropriately protected since these two snapper species are target species of commercial catch and recreational fishing in the region. Thus, understanding precise ecological characteristics of fish aggregations of the two snapper species would be useful for effective management including the necessary spatial scale and duration of spawning ground protection.

Figure 1 Study site and fish aggregations of Lutjanus decussatus and L. fulviflamma.

Location of Yaeyama Islands (A), Sekisei lagoon (B) (enclosed area by a yellow dotted line), fish aggregations of L. decussatus (C) and L. fulviflamma (D). Aerial photograph in (B) was provided by International Coral Reef Research and Monitoring Center.

The aims of the present study were to clarify ecological aspects of the fish aggregations of Lutjanus decussatus and L. fulviflamma. Specifically, the aims were to clarify: (1) seasonality and lunar-phase periodicity of fish aggregation formation, (2) fine-scale spatial distribution of fish density (spatial variations of fish density at intervals of several-tens of meters) at the aggregation site, (3) size and age frequency distributions of fishes in the aggregation site, (4) gonad development of fish individuals that were captured in the aggregation site, (5) to compare fish abundance between inside and outside the aggregation site, and (6) to verify that fish aggregations of the two snapper species were spawning aggregation. Since the present study is the first examination about fish aggregation of the two snapper species, the results would contribute to a more comprehensive understanding of ecological aspects about spawning aggregations of snappers in coral reefs.

Materials and Methods

Main method of this study was field observation. Some fish individuals were caught as samples to examine the gonad development. By placing on ice, these fish individuals were immediately killed to minimize pain. Okinawa prefectural government fisheries coordination regulation No. 37 approved the sampling procedure. This regulation permits capture of fishes for scientific purposes on Okinawan coral reefs.

Study site

This study was conducted at Sekisei lagoon in the Yaeyama Islands, Okinawa, located in the southernmost part of Japan (Fig. 1). One fish aggregation site was located in the Sekisei lagoon, although the precise location is not shown. This is because the study site is not protected during the spawning periods of the two snapper species yet and there is concern that showing the precise location might lead to overfishing the fish aggregations of the two snapper species.

Temporal variations in the fish abundance within the aggregation site

To clarify the monthly and weekly variation in the fish abundance at the aggregation site, daytime underwater observations were conducted. After preliminary surveys between September and October 2020, one line transect (600 m ×5 m) was set to extend over the whole aggregation site between November 2020 and December 2021. Underwater observations on the transect were conducted using SCUBA. During the main spawning season of the two snapper species (between May and October), weekly observations were conducted (Table 1). The observation days were adjusted to be carried out during each of the four lunar phases whenever possible (new moon, first-quarter moon, full moon and last-quarter moon). When rough sea conditions prevented such adjustments, the observation days were set as several days earlier or later of the four lunar phases (Table 1). During non-spawning seasons (between November 2020 and April 2021, and between November and December 2021), underwater observations were conducted around two lunar phases (last-quarter moon and new moon). This was because: (1) preliminary surveys between September and October 2020 revealed the fish aggregations of the two snapper species were found around the last-quarter moon and new moon and (2) Nanami et al. (2010) showed that Lutjanus decussatus showed a greater peak of ovary development during the last-quarter moon.

Table 1 Date of underwater observations and fish sampling (x: conducted).

		Lunar	Underwater	Fish	Number of	samples	
Date	Year	phase	Observation	Sampling	Lutjanus decussatus	Lutjanus fulviflamma	
November 8	2020	LQM	x				
November 15	2020	NewM	x				
December 8	2020	LQM	x				
December 14	2020	NewM-1	x				
January 5	2021	LQM-1	x				
January 13	2021	NewM	x				
February 5	2021	LQM	x				
February 13	2021	NewM+1	x				
March 5	2021	LQM-1	x				
March 13	2021	NewM	x				
April 4	2021	LQM	x	x	N.A.	Male = 1, Female = 1	
April 11	2021	NewM-1	x				
May 3	2021	LQM-1	x	x	Male = 1, Female = 4	N.A.	
May 13	2021	NewM+1	x	x	N.A.	Male = 4, Female = 2	
May 22	2021	FQM+2	x				
May 26	2021	FullM	x				
June 1	2021	LQM-1	x	x	Male = 0, Female = 9	Male = 0, Female = 1	
June 10	2021	NewM	x	x	Male = 0, Female = 2	Male = 8, Female = 2	
June 17	2021	FQM-1	x				
June 25	2021	FullM	x				
July 1	2021	LQM-1	x	x	Male = 5, Female = 10	Male = 3, Female = 6	
July 10	2021	NewM	x	x	Male = 5, Female = 2	Male = 6, Female = 9	
July 17	2021	FQM	x				
July 31	2021	LQM	x	x	Male = 5, Female = 6	Male = 5, Female = 1	
August 9	2021	NewM+1	x	x	Male = 1, Female = 4	Male = 10, Female = 10	
August 16	2021	FQM	x				
August 24	2021	FullM+2	x				
August 29	2021	LQM-1	x	x	Male = 0, Female = 9	N.A.	
September 7	2021	NewM	x	x	Male = 0, Female = 1	Male = 7, Female = 4	
September 16	2021	FQM+2	x				
September 21	2021	FullM	x				
October 2	2021	LQM+3	x	x	Male = 7, Female = 2	Male = 0, Female = 2	
October 7	2021	NewM+1	x	x	N.A.	Male = 7, Female = 3	
October 15	2021	FQM+2	x				
October 20	2021	FullM	x				
October 29	2021	LQM	x				
November 5	2021	NewM	x				
November 26	2021	LQM-1	x				
December 4	2021	NewM	x				
Notes.

Lunar phases are abbreviated as LQM last-quarter moon

NewM new moon

FQM firstquarter moon

FullM full moon

N.A. not available due to low fish density at the aggregation site or logistic constraint

“+” and “-” mean after and before the lunar phase (e.g. “NewM+1” means 1 day after new moon).

The number of individuals of the two snapper species on the above-mentioned 600 m ×5 m line transect was counted per every 1 min. During the observations, a portable GPS receiver, sealed in a waterproof case and attached to a buoy, was towed and the course and distance of the tracks were obtained in accordance with the protocol detailed in Nanami et al. (2017). The water depth range at which the underwater observations were conducted was c.a. 10 m to 15 m.

Analysis of periodicity in the increase of fish abundance in relation to lunar phase

To determine the statistical significance in periodicity of fish increasing at the aggregation site, a time-series analysis (correlogram) was applied. In this procedure, data obtained between May 3 and October 20 was used for the analysis (Table S1), since greater fish abundance at the aggregation site was found in the study period (see RESULTS). In the analysis, 24 time periods were established, in which each time period includes one lunar phase (last-quarter moon, new moon, first-quarter moon and full moon) (Table S1). Correlogram was applied (Fig. S1, Table S2) by using R statistical computing language (function “acf”: R Core Team, 2022). Statistical significance of auto-correlation coefficient was determined by using 95% confidence intervals (CI) as follow: 95%CI=1.96/T

where T is the number of observations.

It can be regarded that the auto-correlation coefficient is significant when the following equation was found: |pk|>95%CI=1.96T

where pk is the value of auto-correlation at k th time lag, and — pk— is the absolute value of pk. If —pk— is greater than the 95% CI, auto-correlation at k th time lag was significant. Since T was 24 in the analysis (Table S1), 95% CI was calculated as 1.96/√ (24) = 0.40. For example, auto-correlation coefficient at 4th time lag (p4) was over 0.40, the p4 is a significantly positive value. Namely, the periodic increase in fish abundance was found at the same particular lunar phase.

Since data for July 22 (full moon phase) could not be collected due to a typhoon, the missing value was imputed. In this procedure, other fish abundance data obtained around full moon phase (May 26, June 25, August 24, September 21 and October 20) was averaged and the average value was applied to impute the missing value.

Fine-scale spatial variations in fish density at the aggregation site

The above-mentioned underwater observations revealed that fish abundance showed positive peaks around last-quarter moon or new moon (see RESULTS). At the two lunar phases, fine-scale spatial variations in fish abundance (variations in fish abundance at intervals of several-tens meters) within the aggregation site were examined. The above-mentioned 600 m × 5 m line transect was divided into 1-minute sub-transects (average distance ± standard deviation = 20.5 m ± 4.1). Then, the number of individuals of the two snapper species and the distance for the 1-minute sub-transect were obtained. From the data, the number of individuals was converted to density (per 20 m × 5 m) for each 1-minute transect. Fish density on the 1-minute sub-transect was individually plotted by bubble plot along the 600 m × 5 m line transect.

The fine-scale spatial variation in fish density per 20 m × 5 m was shown as frequency data (histogram). Then, Kolmogorov–Smirnov test was applied to test the significant difference in fine-scale spatial variation in fish density between the two fish species.

Verification of spawning by ovarian development

Domeier (2012) and de Mitcheson & Erisman (2012) have indicated that presence of females with matured eggs in the hydrated stage (hydrated eggs) in aggregation sites is one of the direct evidence that the fish aggregation can be regarded as a spawning aggregation. To clarify whether females have hydrated eggs in the aggregation site, individuals of the two species were caught by spear-gun just after the above-mentioned underwater observations around the last-quarter and new moon (Table 1). In total, 73 and 92 individuals were caught for L. decussatus and L. fulviflamma, respectively (Table 1). In the laboratory, fork length (FL), whole body weight and gonad weight were measured. The gonadsomatic index (GSI) was calculated by using the formula: GSI=Gonad weight (g)/whole body weight (g)−gonad weight (g)×100.

To obtain histological observations, the gonads were preserved in 20% buffered formalin over 48 h and then kept in 70% ethanol baths. Embedded pieces of gonads were sectioned and stained with Mayer’s hematoxylin–eosin. Under microscopic observations, developmental stages of ovaries were examined whether the gonads were sufficiently developed for spawning (hydrated eggs). The categorization of ovarian developmental stages followed Ohta & Ebisawa (2015) and Ohta et al. (2017). According to the categorization of Ohta & Ebisawa (2015), oocytes with migration nuclear stage, pre-maturation stage and maturation stage were defined as hydrated stage.

Size and age frequency distributions

To determine the size frequency distribution of fishes at the aggregation site, histograms of fork length for the above-mentioned fish samples was plotted. Male and female were separately plotted, and probability density of size frequency was analyzed by R statistical computing language (function “density”: R Core Team, 2022).

To determine the age frequency distribution of fishes, sagittal otoliths of the above-mentioned fish samples were extracted from each fish, and cleaned in water and dried. Then, one otolith was embedded in epoxy resin and transversely sectioned into 0.5 mm-thick sections using ISOMET low speed saw (Buehler) and attached on a glass slide. The sectioned otoliths were observed under a microscope with reflected light at ×4 magnification, and the number of opaque rings was counted. The procedure of opaque rings count followed Ohta et al. (2017). The number of opaque rings on each otolith was counted twice. If the two counts coincided, the number of rings was used. However, if the two counts did not coincide, the number of opaque rings was counted once more and any two coinciding counts were used. Since Nanami (2021) and Shimose & Nanami (2015) revealed that each opaque ring was formed annually for the two species, number of opaque rings can be considered as age (year). Age frequency distribution for male and female was separately plotted, and probability density was analyzed by R statistical computing language (function “density”: R Core Team, 2022).

Mann–Whitney U-test test was applied to determine the significant differences in average fork length and average age between males and females.

Comparison of fish density between inside and outside the aggregation site

Domeier (2012) has proposed the definition of spawning aggregations, indicating that fish abundance in aggregation sites is at least 4-fold greater than that outside the aggregation site. In order to verify the definitions, 65 study sites outside the aggregation site were established in Sekisei Lagoon (Fig. S2) and the number of the two snapper species was counted. Above-mentioned 20-minutes underwater survey with a portable GPS receiver was conducted at each site (details about method was shown in Nanami (2020). Fish abundance per 600 m ×5 m was estimated by using the fish count data and the measured distance. The estimated fish abundance per 600 m ×5 m among the 65 sites were averaged and regarded as average fish abundance outside the aggregation site. Then, the fish abundance per 600 m ×5 m inside the aggregation site was compared with that outside aggregation site.

Results

Temporal variations in the fish abundance and reproductive activity in relation to lunar phase

Greater fish abundance (number of individuals per 600 m ×5 m) of Lutjanus decussatus was found between May and October (Fig. 2A), when the water temperature exceeded 25 °C. During the six months, clear peaks in the fish abundance were only found around the last-quarter moon (Fig. 2B). The peak fish abundance ranged from 529 (October) to 1565 (June). Average GSI values ± SD (standard deviation) of females that were caught at the aggregation site was 10.34 ± 4.95 (ranged from 0.42 to 24.89) (Fig. 2C). About 69% of individuals (34 out of 49) had hydrated stage oocytes (Figs. 2C–2E, Table S3).

Figure 2 Temporal changes in number of fish individuals and gonad-somatic index for Lutjanus decussatus.

Temporal changes in number of fish individuals with water temperature (A). Data in months enclosed by a dotted line in (A) are re-plotted in (B), showing lunar-related temporal changes in the number of fish individuals since a greater number of individuals were observed during the study period. Temporal changes in gonad-somatic index between April and October (C), together with oocyte developmental stage (pie charts). Numerals in pie charts indicate the number of females with eggs in each developmental stage. Lunar phases are abbreviated as LQM, last-quarter moon; NewM, New moon. “+” and “-” mean after and before the lunar phase (e.g., “NewM-1” means 1 day before new moon). Oocyte developmental stage was abbreviated as PYS, primary yolk stage; TYS, tertiary yolk stage; MN, migration nuclear stage; PMA, pre-maturation stage; MA, maturation. Fish individual with hydrated egg (D) and cross-section of ovaries having oocytes with maturation stage (E). In (E), MA represents maturation stage of oocytes. Data of water temperature in (A) was provided by International Coral Reef Research and Monitoring Center.

Greater fish abundance of L. fulviflamma was clearly found between April and October (Fig. 3A), when water the temperature exceeded 25 °C. During the seven months, clear peaks in the fish abundance were found around the last-quarter moon in April, May, June and October (Fig. 3B). In contrast, clear peaks were found around the new moon in July, August and September (Fig. 3B). The peak fish abundance ranged from 660 (September) to 6337 (June). Average GSI values ± SD of females were 6.33 ± 2.21 (ranged from 3.44 to 14.51) (Fig. 3C). About 90% of individuals (37 out of 41) had hydrated stage oocytes (Figs. 3C–3E, Table S4).

Figure 3 Temporal changes in number of fish individuals and gonad-somatic index for Lutjanus fulviflamma.

Temporal changes in number of fish individuals with water temperature (A). Data in months enclosed by a dotted line in (A) are re-plotted in (B), showing lunar-related temporal changes in number of fish individuals since greater number of individuals were observed during the study period. Temporal changes in gonad-somatic index between April and October (C), together with oocyte developmental stage (pie charts). Numerals in pie charts indicate the number of females with eggs in each developmental stage. Lunar phases are abbreviated as LQM, last-quarter moon; NewM, New moon. “+” and “-” mean after and before the lunar phase (e.g., “NewM-1” means 1 day before new moon). Oocyte developmental stage was abbreviated as TYS, tertiary yolk stage; MN, migration nuclear stage; PMA, pre-maturation stage; MA, maturation. Fish individual with hydrated egg (D) and cross-section of ovaries having oocytes with maturation stage (E). In (E), MA represents maturation stage of oocytes. Data of water temperature in (A) was provided by International Coral Reef Research and Monitoring Center.

Periodicity of the increase of fish density

Correlogram of L. decussatus revealed that significant positive auto-correlation coefficients were found when time lags were 4 and 8 (Fig. 4A). Correlogram of L. fulviflamma revealed that significant positive auto-correlation coefficient was found when time lag was 4 (Fig. 4B).

Figure 4 Correlogram showing auto-correlation coefficient with time lag, which examines the periodicity in fish increasing or decreasing at a particular time lag (for time lag, see Tables S1, S2 and Fig. S1).

Horizontal dotted line represent 95% confidence interval of the auto-correlation coefficient. Black and white arrows represent significant positive and negative values of auto-correlation coefficient, showing significant periodicity at the time lag.

Fine-scale spatial variations in fish density at the aggregation site

Fine-scale spatial distributions revealed that L. decussatus showed relatively uniform distribution at the aggregation site, especially around the last-quarter moon between May and October (Fig. 5). Fish densities per 20 m ×5 m were generally less than 140 individuals in most cases (Fig. 6).

Figure 5 Spatial distribution of Lutjanus decussatus on the 600 m × 5 m line transect in the aggregation site.

The transect was divided into 1-minute sub-transects. Fish data are shown as bubble plots and each bubble represents the fish density (number of individuals in 20 m × 5 m area) on each 1-minute sub-transect. Cross marks represent no fishes in the sub-transect. Graphs at upper right side show the temporal changes in number of fish individuals (see Fig. 2A) and red arrows represent the day when the data was collected. Lunar phases are abbreviated as LQM, last-quarter moon; NewM, New moon. “+” and “-” mean after and before the lunar phase (e.g., “NewM-1” means 1 day before new moon).

Figure 6 Fish density frequency of Lutjanus decussatus for 1-minute sub-transect in the aggregation site.

Fish density represents the number of fish individuals in the 20 m × 5 m area. Number of individuals in the 1-minute sub-transect was converted to fish density (20 m × 5 m). Graphs at upper right side show the temporal changes in number of fish individuals (see Fig. 2A) and red arrows represent the day when the data was collected. Lunar phases are abbreviated as LQM, last-quarter moon; NewM, New moon. “+” and “-” mean after and before the lunar phase (e.g., “NewM-1” means 1 day before new moon). For actual spatial distributions, see Fig. 5.

In contrast, greater fish density of L. fulviflamma per 20 m ×5 m was found at particular areas (south-eastern side) in the aggregation site (Fig. 7). This tendency was clearly found around last quarter moon in April, May and June, and around new moon in July and August. In some cases, fish density within 20 m ×5 m was over 300 (Fig. 8).

Figure 7 Spatial distribution of Lutjanus fulviflamma on the 600 m × 5 m line transect in the aggregation site.

The transect was divided into 1-minute sub-transects. Fish data are shown as bubble plots and each bubble represents the fish density (number of individuals in 20 m × 5 m area) on each 1-minute sub-transect. Cross marks represent no fishes in the sub-transect. Graphs at upper right side show the temporal changes in number of fish individuals (see Fig. 3A) and red arrows represent the day when the data was collected. Lunar phases are abbreviated as LQM, last-quarter moon; NewM, New moon. “+” and “-” mean after and before the lunar phase (e.g., “NewM-1” means 1 day before new moon).

Kolmogorov–Smirnov test revealed that a significant difference in fish density frequency (i.e., patterns in fine-scale spatial distribution) was found between the two fish species from May to October (p < 0.05, Table 2).

Figure 8 Fish density frequency of Lutjanus decussatus for 1-minute sub-transect in the aggregation site.

Fish density represents the number of fish individuals in the 20 m × 5 m area. Number of individuals at 1-minute sub-transect was converted to fish density (20 m × 5 m). Graphs at upper right side show the temporal changes in number of fish individuals (see Fig. 3A) and red arrows represent the day when the data was collected. Lunar phases are abbreviated as LQM: last-quarter moon; NewM: New moon. “+” and “-” mean after and before the lunar phase (e.g., “NewM-1” means 1 day before new moon). For actual spatial distributions, see Fig. 7.

Size and age frequency distributions

For L. decussatus, fork length of most individuals was between 200.0–279.5 mm for both males and females (Figs. 9A, 9B). Average fork length of females (243.9 mm FL ± 21.4 SD: standard deviation) was significantly greater than males (229.2 mm FL ± 19.6 SD: Mann–Whitney U-test, p < 0.01). Highest probability of occurrence in fish length was about 210 mm FL for males whereas about 260 mm FL for females. Although average age was not significantly different between males (9.4 years ± 5.0 SD) and females (8.1 years ± 4.4 SD: Mann–Whitney U- test, p = 0.26), highest probability of occurrence in age was about 6 years for both males and females (Figs. 9C, 9D).

For L. fulviflamma, fork length of most individuals was less than 250 mm for males whereas 280 mm for females (Figs. 10A, 10B). Average fork length of females (246.9 mm FL ± 23.0 SD) was significantly greater than males (233.9 mm FL ± 17.8 SD: Mann–Whitney U-test, p < 0.01). Probability density revealed that highest probability of occurrence in fish length was about 240 mm FL for males whereas about 260 mm FL for females. Although average age was not significantly different between males (6.8 years ± 3.1 SD) and females (8.1 years ± 4.5 SD: Mann–Whitney U-test, p = 0.29), highest probability of occurrence in age was about 7 years and 5 years for males and females, respectively (Figs. 10C, 10D).

Comparison of fish density between inside and outside the aggregation site

Fish abundance per 600 m ×5 m of L. decussatus inside the aggregation site was 148.2 to 438.4-fold (average = 266.8-fold) greater than that outside the aggregation site (Table 3).

Fish abundance of L. fulviflamma inside the aggregation site was 33000.0 to 316850.0-fold (average = 141557.1-fold) greater than that outside the aggregation site (Table 3).

Discussion

Verification of spawning aggregation of two snapper species

For the two snapper species Lutjanus decussatus and L. fulviflamma, this study was the first attempt to examine whether fish aggregations at a particular site can be regarded as spawning aggregations. The present study showed: (1) repeated greater fish abundance of the two species and this is predictable in time (particular month and lunar-phase) and space, (2) the fish abundance inside the aggregation site is over 4-fold greater than that outside the aggregation site, and (3) most females inside the aggregation site had hydrated eggs. Thus, the fish aggregations of the two snapper species can be regarded as spawning aggregations.

Spawning season and spawning day

The present study revealed that fish aggregations of the species were found between May and October. Nanami et al. (2010) showed similar results from fish samples by commercial catch, i.e., estimated main spawning season of L. decussatus was between June and October. Since developed- or matured-oocytes (tertiary yolk stage, migration nuclear stage, pre-maturation stage and maturation stage) were obtained for most female individuals between May and October, it is suggested that the main spawning season of L. decussatus can be regarded as between May and October in the study region. Since peaks of fish abundance were found only around the last-quarter moon and the periodicity in the increase of fish abundance during each last-quarter moon phase was significant, it is suggested that spawning occurs around the last-quarter moon.

The present study revealed that spawning aggregations of the species were found and the most females had hydrated eggs between April and October. Shimose & Nanami (2015) showed similar results from fish samples by commercial catch, i.e., estimated main spawning season of L. fulviflamma is between April and August. Therefore, it is suggested that the spawning season of L. fulviflamma can be regarded as between April and October in the study region. The peak fish abundance at the spawning ground was found during the last-quarter moon (in April, May, June and October) and new moon (in July, August and September), suggesting that spawning occurs around the last-quarter moon and new moon. This lunar-related periodicity in the increase of fish abundance at during a particular lunar phase (last-quarter moon or new moon) was also supported by the correlogram in the present study. This evidence for lunar-related spawning is the first finding for L. fulviflamma.

Since western Atlantic snapper species (Lutjanus cyanopterus (Cuvier, 1828) and L. jocu (Bloch & Schneider, 1801)) spawned for a period of four to seven consecutive days (Heyman & Kjerfve, 2008), the two snapper species in this study might spawn during several consecutive days around last-quarter moon and new moon. Intensive daily observations around the two lunar phases would clarify more precise ecological aspects about aggregation formation of the two snapper species.

Table 2 Results of Kolmogorov-Smirnov test to test the significant difference in fish density (number of fish in 20 m ×5 m area) frequency at aggregation site between Lutjanus decussatus and L. fulviflamma.

	Lunar		n	n	
Date	phase	p-value	(L. decussatus)	(L. fulviflamma)	
April 4	LQM	0.155	25	25	
April 11	NewM-1	>0.999	33	33	
May 3	LQM-1	<0.001	36	36	
May 13	NewM+1	0.124	23	23	
June 1	LQM-1	<0.001	37	37	
June 10	New M	0.035	30	30	
July 1	LQM-1	0.001	40	40	
July 10	NewM	0.056	28	28	
July 31	LQM	<0.001	29	29	
August 9	NewM+1	0.012	28	28	
August 29	LQM-1	<0.001	30	30	
September 7	NewM	0.007	35	35	
October 2	LQM+3	0.004	27	27	
October 7	NewM+1	0.037	25	25	
Notes.

Lunar phases are abbreviated as LQM last-quarter moon

NewM new moon

“+” and “-” mean after and before the lunar phase (e.g. “NewM+1” means 1 day after new moon).

Figure 9 (A–D) Size and age frequency of Lutjanus decussatus individuals that were captured at the aggregation site.

Solid lines represent the probability density function. *: Among the 49 individuals in (B) age of one individual could not be identified due to difficulty in counting of number of opaque rings on otolith. Thus, sample size was 48 in (D).

Figure 10 (A–D) Size and age frequency of Lutjanus fulviflamma individuals that were captured at the aggregation site.

Solid lines represent the probability density function.

Water temperature is one of the main factors for gonad development and reproduction of marine fishes (Wang et al., 2010). The present study revealed that reproductive activity of the two snapper species is likely to occur when water temperature is over 25 °C. In addition, decline of temperature (beginning of October) might be one of the factors leading to reduced ovarian development. This trend is consistent with cubera snapper (Lutjanus cyanopterus) in the Caribbean (Heyman et al., 2005; Motta et al., 2022), showing that spawning aggregation of L. cyanopterus related to increasing water temperature in the summer.

Fine-scale spatial variation in fish density within spawning ground

Some previous studies have shown fine-scale spatial variations of fish density at intervals of several-tens meters within spawning grounds for groupers (e.g., Colin, 2012; Nanami et al., 2017; Sadovy de Mitcheson et al., 2020). These studies have shown species-specific spatial variations among multiple grouper species and each species showed species-specific core sites, in which a very high density was found in a limited area within the spawning ground (Nanami et al., 2017; Sadovy de Mitcheson et al., 2020). Spatial variations of fish density at intervals of 250 m was also observed for two snapper species (Biggs & Nemeth, 2016), showing fine scale movement of two snapper species (L. cyanopterus and Lutjanus jocu) within the aggregation site in relation to spawning time.

Table 3 Comparison about number of fish individuals per 600 m ×5 m of two snapper species between inside and outside aggregation site.

Average number of fish individuals outside the aggregation site were obtained from underwater observations at 65 sites (Fig. S2).

	Inside	aggregation	site		Outside aggregation site		
	Month	Lunar phase	Number of fish individuals		Number of fish individuals	Inside/ Outside	
Lutjanus decussatus	May 3	LQM-1	903		3.57	252.9	
	June 1	LQM-1	1565		3.57	438.4	
	July 1	LQM-1	877		3.57	245.7	
	July 31	LQM	939		3.57	263.0	
	August 29	LQM-1	901		3.57	252.4	
	October 2	LQM+3	529		3.57	148.2	
	Average		952.33		3.57	266.8	
Lutjanus fulviflamma	April 4	LQM	1521		0.02	76050.0	
	May 3	LQM-1	4040		0.02	202000.0	
	June 1	LQM-1	6337		0.02	316850.0	
	July 10	NewM	3000		0.02	150000.0	
	August 9	NewM+1	3495		0.02	174750.0	
	September 7	NewM	660		0.02	33000.0	
	October 2	LQM+3	765		0.02	38250.0	
	Average		2831.14		0.02	141557.1	
Notes.

Lunar phases are abbreviated as LQM last-quarter moon

NewM new moon

“+” and “-” mean after and before the lunar phase (e.g. “NewM+1” means 1 day after new moon).

The present study revealed a significant difference in the fine-scale spatial distributions in fish density between the two snapper species. Although the reasons why the two species showed species-specific spatial variations within the spawning ground remain unknown, this might be related with preference to a particular environmental condition or mating behavior.

Size and age frequency of fish aggregation

The present study showed that the maximum fork length of fish individuals at the spawning ground was 304 mm and 295.5 mm for L. decussatus and L. fulviflamma, respectively. In contrast, previous studies revealed that maximum fork length was 316.5 mm and 347.0 mm for Lutjanus decussatus and L. fulviflamma, respectively (Shimose & Nanami, 2015; Nanami, 2021). Since average fork length of fish individuals at the spawning ground was less than 250 mm for both species, most of the fish individuals forming the spawning aggregations were relatively small sized individuals.

A similar trend was also found for age frequency. Maximum age was respectively 24 and 23 for Lutjanus decussatus and L. fulviflamma (Shimose & Nanami, 2015; Nanami, 2021), whereas maximum age of fish individuals at the spawning ground was respectively 21 and 17 for L. decussatus and L. fulviflamma. Since average age of fish individuals at the spawning ground was less than 10 for both species, most of the fish individuals that form the spawning aggregations were relatively young individuals.

Conclusions

Results of the present study revealed that fish aggregations of the two snapper species (Lutjanus decussatus and L. fulviflamma) can be regarded as spawning aggregations due to their spatially and temporally predictable formation of fish aggregation as well as the presence of hydrated eggs in females within the aggregation site. In the Caribbean and Palau, spawning aggregations for some species of snappers have been previously found by local communities (Hamilton, Sadovy de Mitcheson & Aguilar-Perera, 2012). Since there is almost no local ecological information about the spawning aggregations of L. decussatus and L. fulviflamma, the present study is probably the first finding of a spawning aggregation of the two snapper species. As the two species are major fishery target species in the study region, the results of the present study should be applied to consider when and where a marine protected area should be established for effective protection of the spawning aggregation of the two species. Namely, the aggregation site of the two snapper species should be protected during their main spawning season (between April and October) and the protected area should completely cover the fish aggregations.

Supplemental Information

Supplemental Information 1 Spawning aggregation of Lutjanus decussatus

Click here for additional data file.

Supplemental Information 2 Spawning aggregation of Lutjanus fulviflamma

Click here for additional data file.

Supplemental Information 3 Details about the “time period” settings for the time-series analysis (correlogram)

Twenty-four time periods were set based on four lunar phases (last-quarter moon, new moon, first-quarter moon and full moon). The observation days were adjusted to the four lunar phases whenever possible. When rough sea condition prevented from the adjustment, the observation days were set as several days earlier or later of the four lunar phases (see ‘Materials and methods’). Lunar phases are abbreviated as LQM, last-quarter moon; NewM, new moon; FQM, first-quarter moon; FullM, full moon. “+” and “-” mean after and before the lunar phase (e.g., “NewM+1” means 1 day after new moon). *: underwater observation was not be conducted due to typhoon.

Click here for additional data file.

Supplemental Information 4 Detail about “time lag”, “time period” and lunar phase for a time-series analysis (correlogram)

Numbers (1 - 24) represent “time period” (see Table S1). Twenty-four time periods were set based on four lunar phases (last-quarter moon, new moon, first-quarter moon and full moon). Lunar phases are abbreviated as LQM, last-quarter moon; NewM, new moon; FQM, first-quarter moon; FullM, full moon. For example, “1-LQM” means the fish density data was collected at time period “1” and the lunar phase was at the time period was around last-quarter moon. Time lag = 0, 4, 8, 12: auto-correlation coefficients were calculated by using fish density data that were collected at the same lunar phases. Time lag = 1, 2, 3, 5, 6, 7, 9, 10, 11: auto-correlation coefficients were calculated by using fish density data that were collected at different lunar phases.

Click here for additional data file.

Supplemental Information 5 Size, gonadsomatic index (GSI) and ovary developmental stage in relation to sampling date for Lutjanus decussatus

Ovary developmental stages are shown as abbreviations: PYS, primary yolk stage; TYS, tertiary yolk stage; MNS, migration nuclear stage; PMA, pre-maturation stage; MA, maturation stage. Lunar-phase are abbreviated as LQM, last-quarter moon; NewM, new moon. “+” and “-” mean after and before the lunar phase (e.g., “NewM+1” means 1 day after new moon).

Click here for additional data file.

Supplemental Information 6 Size, gonadsomatic index (GSI) and ovary developmental stage in relation to sampling date for Lutjanus fulviflamma

Ovary developmetal stages are shown as abbreviations: TYS, tertiary yolk stage; MNS, migration nuclear stage; PMA, pre-maturation stage; MA, maturation oocytes. Lunar-phase are abbreviated as LQM, last-quarter moon; NewM: new moon. “+” and “-” mean after and before the lunar phase (e.g., “NewM+1” means 1 day after new moon).

Click here for additional data file.

Supplemental Information 7 Schematic explanation how to analyze the periodicity of fish density increase by correlogram (this figure is not real data)

A time-series of data (24 data plots that were obtained at 24 time periods) is shown (for details about time period setting, see Table S1). Time lag was 1 when the original data (blue-colored data) was shift a 1-time period. Then, auto-correlation coefficient between original data and shifted data (magenta-colored data) was calculated for each time lag (A–E). By using numerical value of the auto-correlation coefficient for each time lag, correlogram (F) was made. This procedure was repeated until time lag reached 12 (see Table S1).

Click here for additional data file.

Supplemental Information 8 Location of the 65 study sites to clarify the fish density of Lutjanus decussatus and L. fulviflamma outside the aggregation site

Overall study area (A) and location of each study site (B). Aerial photographs were provided by International Coral Reef Research and Monitoring Center.

Click here for additional data file.

Supplemental Information 9 Figure 2 raw data.

Click here for additional data file.

Supplemental Information 10 Figure 3 raw data.

Click here for additional data file.

Supplemental Information 11 Figure 4 raw data.

Click here for additional data file.

Supplemental Information 12 Figure 5 raw data.

Click here for additional data file.

Supplemental Information 13 Figure 6 raw data.

Click here for additional data file.

Supplemental Information 14 Figure 7 raw data.

Click here for additional data file.

Supplemental Information 15 Figure 8 raw data.

Click here for additional data file.

Supplemental Information 16 Figure 9 raw data.

Click here for additional data file.

Supplemental Information 17 Figure 10 raw data.

Click here for additional data file.

Supplemental Information 18 Table 2 raw data.

Click here for additional data file.

Supplemental Information 19 Table 3 raw data.

Click here for additional data file.

I express my grateful thanks to M. Sunagawa and S. Sunagawa for piloting the research boat YAEYAMA, K. Aramoto, M. Yoshida, F. Nakamura and K. Ohishi for their assistance in the field, K. Nema for assistance in laboratory works and the staff of Yaeyama Field Station, Fisheries Technology Institute for support during the present study. I would like to thank Dr. Norman for English language editing. Providing of aerial photograph and data of water temperature from International Coral Reef Research and Monitoring Center and constructive comments from K. Mansor and two anonymous reviewers were much appreciated. The present study complies with current laws in Japan.

Additional Information and Declarations

Competing Interests

Author Contributions

Animal Ethics

Data Deposition

The author declares that there are no competing interests.

Atsushi Nanami conceived and designed the experiments, performed the experiments, analyzed the data, prepared figures and/or tables, authored or reviewed drafts of the article, and approved the final draft.

The following information was supplied relating to ethical approvals (i.e., approving body and any reference numbers):

Fisheries coordination regulation no. 37 by Okinawa Prefectural Government.

This study mainly involved observing free-living fishes in their natural habitat. Individuals that were caught by spearing for sampling were immediately killed by placing them on ice to minimize pain. The sampling procedure was approved by the Okinawa Prefectural Government in compliance with the fisheries coordination regulation no. 37, which permits the capture of marine organisms on Okinawan coral reefs for scientific purposes.

A copy of the fisheries coodination regulation is attached as a Supplementary File.

In addition, English translation for regulation No. 37 is also attached as a Supplementary File.

The following information was supplied regarding data availability:

The raw measurements are available in the Supplementary File. The location of the study site is confidential.

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
