# Peer review of "Spawning aggregations of checkered snapper (Lutjanus decussatus) and blackspot snapper (L. fulviflamma): seasonality, lunar-phase periodicity and spatial distribution within spawning ground"

_PeerJ, doi:10.7717/peerj.15991_

## Round 0.1 · original submission · Major Revisions

The data collected are very important and could be very useful. However, the manuscript could be greatly improved. In fact, the major criticism is related to the metric used (fish density), which is not suitable for the objective. Instead, it would be better to use fish biomass (g.m-2) for a better understanding.
I invite you to revise and resubmit your manuscript.
Best regards,

Reviewer 1 ·

Basic reporting

The article is extremely well written, clear and unambiguous.

Experimental design

no comment.

Validity of the findings

no comment.

Additional comments

The paper is very well done and could be published as is.

Nuances and nitpicks:

Line 367. Fine scale movement of snapper on an aggregation site was described and could be referenced (Biggs and Nemeth 2016)

Biggs, C. R., and R. S. Nemeth. 2016. Spatial and temporal movement patterns of two snapper species at a multi-species spawning aggregation. Marine Ecology Progress Series 558:129-142.

Line 359. Snappers in the wider Caribbean tend to spawn in “spring time” while waters are warming towards the summer. This may be a trigger in addition to the threshold temperature 25 degrees. The word increased could be changed to increasing

The authors should consider that sampling occurred at intervals of seven days. Western Atlantic snappers often spawn for a period of 4 - 7 consecutive days. The conclusions of this paper would not change if sampling occurred daily but the description of peak spawning period could be described in more detail.

slide glass should be glass slide.

·

Basic reporting

Review of the MS titled Spawning aggregations of checkered snapper (Lutjanus decussatus) and blackspot snapper (L. fulviflamma): seasonality, lunar-phase periodicity and spatial distribution within spawning ground”

The author have provided a thorough and detailed investigation into the ecological characteristics of fish aggregations of two snapper species (L. decussatus and L. fulviflamma), including seasonality and lunar-phase periodicity, spatial distribution, size and age frequency distributions, gonad development, and fish density comparisons inside and outside the aggregation site. The use of underwater observations using a 600m x 5m transect was a reliable method for collecting data on fish density and distribution within the aggregation site. The author have also provided clear documentation of their results, including the use of correlograms to confirm lunar-related periodicity in the increase of fish density. The author provided raw data for their statistical analysis and diagram representations.
Overall, the study provides valuable insights into the spawning aggregation behavior of these important fisheries target species, and the results have implications for fisheries management and conservation efforts.
Overall, the study is to well-written and well-structured, with professionally-exposed results and accurate statistical analyses.
Minor comments for the author include avoiding subheadings in the discussion section and ensuring that they respect the zoological code for species authorships in each section. Add standard deviation for the averages of fork length and age of males and females for L. fulviflamma and L. decussatus

Experimental design

The experimental design of this study is well-defined and relevant to the aims and scope of the journal. The research question is clearly stated, and the author has identified a knowledge gap in the ecological characteristics of fish aggregations of two snapper species. The investigation appears to be rigorous and performed to high technical standards.

The methods used in the study are described with sufficient detail and information to replicate the experiment. The use of underwater observations and time-series analysis to investigate the seasonality and lunar-phase periodicity of fish aggregation formation, as well as the fine-scale spatial distribution of fish density, is appropriate for the research question. The author has also provided information on the size and age frequency distributions of fishes in the aggregation site, gonad development, and fish density comparisons inside and outside the aggregation site, which adds depth to the investigation.

Overall, the experimental design of this study appears to be sound and original, with a clear research question, rigorous investigation, and appropriate methods. The study fills an identified knowledge gap in the ecological characteristics of fish aggregations of two snapper species and provides valuable insights for fisheries management and conservation efforts.

Validity of the findings

The findings of this study appear to sound. The author has presented well-controlled, statistically valid data regarding the ecological characteristics of two snapper species' fish aggregations. Furthermore, the author has provided sufficient detail for others to replicate their experiment.

The conclusions of the study are clear, directly related to the original research question, and supported by the results. The author has demonstrated that these two snapper species' fish aggregation formations occur predictably in certain months and lunar phases, with fish densities within the aggregation site significantly higher than outside. Additionally, the study shows that these aggregations serve as reproduction sites for the two species, which has important implications for conservation and fisheries management efforts.

Additional comments

It is worth noting that the impact has not been evaluated, so I recommend the author to add in the conclusion the ecological implication of this study and the management strategy.

Reviewer 3 ·

Basic reporting

The English language can be improved.

Some literature references can be updated.

Experimental design

I disagree with the metric used of fish density, which uses only fish lenght. I suggest use alternatively the fish biomass metric (g.m-2) for a better comphrehesion at all.

Validity of the findings

The findings are very important and highly apllied to management and conservation of the species.

Additional comments

All my comments are found in detail in the pdf (at attach).

Annotated reviews are not available for download in order to protect the identity of reviewers who chose to remain anonymous.

---

## Round 0.2 · accepted · Accept

I am pleased to confirm that your paper has been accepted for publication in PeerJ.

Thank you for submitting your work to this journal.

With kind regards,

Reviewer 3 ·

Basic reporting

Congrats for the detailed revision. The manuscrit greatly improved. I have not nore contributions to the manuscript and I believe that it is now ready for publication.

Experimental design

No comment

Validity of the findings

No comment

Additional comments

No comment